# Lightweight RepVGG-Based Cross-Modality Data Prediction Method for Solid Rocket Motors

**DOI:** 10.3390/s23229165

**Published:** 2023-11-14

**Authors:** Huixin Yang, Shangshang Zheng, Xu Wang, Mingze Xu, Xiang Li

**Affiliations:** 1School of Aerospace Engineering, Shenyang Aerospace University, Shenyang 110136, China; yanghuixin2014@163.com (H.Y.); zhengshangshang@stu.sau.edu.cn (S.Z.); wangxu2@stu.sau.edu.cn (X.W.); xumingze@stu.sau.edu.cn (M.X.); 2Key Laboratory of Education Ministry for Modern Design and Rotor-Bearing System, Xi’an Jiaotong University, Xi’an 710049, China

**Keywords:** solid rocket motor, cross-modality data prediction method, pressure, thrust

## Abstract

Solid rocket motors (SRMs) have been popularly used in the current aerospace industry. Performance indicators, such as pressure and thrust, are of great importance for rocket monitoring and design. However, the measurement of such signals requires high economic and time costs. In many practical situations, the thrust measurement error is large and requires manual correction. In order to address this challenging problem, a lightweight RepVGG-based cross-modality data prediction method is proposed for SRMs. An end-to-end data prediction framework is established by transforming data across different modalities. A novel RepVGG deep neural network architecture is built, which is able to automatically learn features from raw data and predict new time-series data of different modalities. The effectiveness of the proposed method is extensively validated with the field SRM data. The accurate prediction of the thrust data can be achieved by exploring the pressure data. After calculation, the percentage error between the predicted data and the actual data is less than 5%. The proposed method offers a promising tool for cross-modality data prediction in real aerospace industries for SRMs.

## 1. Introduction

Solid rocket motors have been popularly used in the current aerospace industry. With their high specific impulse, simple structure, and safety features, SRMs have become a widely used propulsion system in missiles and other military weapons. The basic components of an SRM include combustion chambers, nozzles, and igniters, with the propellant’s thermal energy converted into kinetic energy to generate thrust [1,2]. Achieving high-efficiency and quality SRMs is essential in ensuring flight safety and strike accuracy [3]. Thrust data are essential for supporting subsequent SRM experiments [4,5]. However, the traditional method of measuring thrust data is associated with high time and economic costs, and it often results in significant errors. These limitations hinder the progress of SRM experiments and the development of SRM technology. Therefore, there is a need for a simple and effective method to measure thrust data for SRM performance studies.

During the testing of solid rocket motors, the traditional method for measuring thrust involves installing the motor on a reaction platform, initiating the motor for testing, and recording the force exerted on the platform [6]. According to Newton’s third law, the thrust generated by the motor is transmitted to the platform, resulting in a reaction force on the platform that is equal in magnitude but opposite in direction to the motor’s thrust. By utilizing sensors installed on the platform, the thrust of the motor can be calculated by measuring the reaction force. However, in practical scenarios, various factors, such as installation errors of individual components and mechanical structure clearances, can introduce inconsistencies in the stress state of the sensors during the experiment compared to the static calibration. This inconsistency can lead to significant errors in the measured thrust results. In the modeling and simulation of solid rocket motors, an alternative approach to obtaining thrust data involves calculating it from pressure data. However, this method typically requires substantial computational power, and it is also time-consuming and expensive. To support the experimental process of motor technology and overcome the limitations and inefficiencies of traditional data measurement methods, a simple and effective approach is required to generate accurate motor thrust data [7,8]. To address this challenge, we propose a lightweight RepVGG-based cross-modality data prediction method for SRM. This novel approach leverages motor pressure data to predict and generate thrust data, which can provide essential support for subsequent motor experiments and performance research. By utilizing deep learning techniques, our method can effectively overcome the errors and limitations associated with traditional measurement methods, resulting in more accurate and reliable thrust data for motor experiments.

Data-driven AI technology has gradually emerged, and deep learning is one of the current representative technologies of AI. In terms of data measurement, some people have proposed algorithms for data measurement and achieved good prediction results, including artificial neural network (NN) [9], convolutional neural network (CNN) [10], etc., with more outstanding learning ability. Deep learning uses the backpropagation algorithm to discover the complex structure of large data sets to indicate how the machine should change its internal parameters, which are used to calculate each layer’s representation from the previous layer’s representation. The deep convolution network has made breakthroughs in image, video, voice, and audio processing, while the recurrent neural networks have made great strides in sequential data, such as text and voice [11]. Applying deep learning methods to the thrust prediction of SRM can establish models through deep learning. The nonlinear, complex, and multidimensional systems in the performance parameters can be solved by the model, bypassing the complex calculation formulas of SRM. The deep learning method achieves better results in the prediction of SRM thrust data.

In recent years, methods for using deep learning to solve SRM problems have received more and more attention. Liu et al. [12] proposed and studied a deep convolutional neural network architecture for defect diagnosis of solid propellants to evaluate the defect scale under the coexistence of internal pore cracks and propellant disbanding. Compared with traditional sequential CNNs, this architecture shows its effectiveness and accuracy in predicting defects. To detect the integrity of aero-motor blades, Wei Feng and colleagues [13] proposed an improved R-CNN network to establish a damage and detection model related to aero-motor blades, with a detection accuracy of 98.81%. Tsutsumi and colleagues [14] successfully used bivariate time series analysis for data-driven fault detection to detect system failures caused by improper valve operation through experiments. Park et al. [15] combined convolutional neural networks and extended short-term memory networks (LSTM) as fault detection for liquid-propellant rocket motors and compared them with traditional methods, confirming the effectiveness of this method. Dai and colleagues [16] proposed an automatic defect analysis system based on the YOLO-V4 model for SRM grain tomography, while Li et al. [17] applied YOLO-V4 to detect internal defects in SRM housing. Gamdha et al. [18] used convolutional neural networks to detect anomalies in solid propellants in rocket motors. Hoffmann et al. [19] used a multilayer perceptron (MLP) to analyze the defect problem of SRM’s propellant interface. At the same time, Guo and colleagues [20] proposed a wavelet packet transformation (WPT) method combined with machine learning, which also achieved good results.

Williams et al. [21] proposed the concept of “virtual sensors”, which combines machine learning with obtained measurement data to provide critical information that the sensor cannot be placed at key locations of rocket motors due to environmental factors. Lv et al. [22] identified the state of liquid rocket motors by constructing a fusion recurrent convolutional neural network (FRCNN). Zavoli et al. [23] used neural networks to predict the performance of hybrid rocket motors. Li et al. [24] attempted to explore contactless event visual data for machine fault diagnosis. Zhang et al. [25] proposed a joint transfer learning method for mechanical fault diagnosis. Dhaked et al. [26] proposed using LSTM to predict the power generation of solar power plants. Zhao and colleagues [27] proposed a convolutional deep belief network to learn the representative features of bearing faults for classification. The experimental results show that the proposed method achieved significant improvements and is more effective than the traditional methods. Li et al. [28] proposed using deep learning methods to diagnose planetary gear faults, and their application to planetary gear datasets verified the superiority of the proposed method. Li et al. [29] used deep learning to assess anxiety levels and provided an automatic assessment technique based on human–computer interaction and virtual reality for mental health physical examination. Wang et al [30] proposed a new intelligent fault diagnosis (IFD) method for planetary gearboxes, which achieved an accuracy of 98.53% in the source mission. Han et al. [31] proposed a deep residual joint transfer strategy method for the cross-condition fault diagnosis of rolling bearings, and successfully identified local faults of motor bearings and planetary bearings.

Deep learning has many applications in mechanical fault diagnosis. Peng et al. [32] proposed a residual mixed-domain attention CNN method to solve the mechanical fault diagnosis problem. Liang et al. [33] proposed a deep convolutional element-learning network to solve the problem of the low generalization performance of bearing fault diagnosis using limited data. Li et al. [34] proposed a mechanical fault diagnosis method based on federated transfer learning to solve the problem of difficulty in collecting data. Zhang et al. [35] proposed an attention-based deep reinforcement learning method to solve the tactical conflict problem of unmanned aerial systems. Zhang et al. [36] proposed an estimation method based on a gradual decreasing current, double correlation analysis and a gated recurrent unit to effectively estimate the state of health of a batteries.

Researchers are increasingly applying deep learning methods to the performance parameters of SRM. The performance parameters of SRM, such as internal ballistic parameters and specific impulses, are essential indicators to measure the performance of SRM. Deep learning has great potential for the calibration of SRM parameters.

Deep learning provides critical information that traditional measurement methods are unable to obtain, reducing the cost of measuring motor parameters. However, the research on the SRM thrust measurement method using deep learning is lacking. Traditional methods for measuring thrust data are often inefficient and have low measurement accuracy, and SRM experiments require a significant amount of data to support them. To address this issue, this study proposes the use of deep learning to predict thrust parameter data for SRMs. We validate the feasibility of this approach by comparing the generated data with actual experimental data obtained from motors. The results demonstrate the effectiveness of this method in generating accurate thrust parameter data, which can serve as a valuable supplement to the ground experiment process of SRMs.

The primary objective of this study is to collect pressure and thrust data from ground experiments of SRMs using sensors and preprocess the obtained data into training and test sets. Subsequently, the collected data will be utilized to develop a predictive model for thrust data, utilizing deep learning techniques. The data processing phase will comprise several essential steps, including filtering, normalization, and feature extraction. The deep learning models will be trained using a training set, and their performance will be evaluated using a test set. The optimal parameters of the model will be determined using experimental comparison methods, and the model’s performance will be assessed using metrics such as mean squared error and mean absolute error. The process is illustrated in Figure 1.

## 2. Deep Learning Method

This section primarily focuses on the system composition and thrust data prediction process of the lightweight one-dimensional RepVGG network. The training process predominantly employs the convolution module within the model to conduct convolution operations on the motor parameter data, thereby capturing data features. The network model’s composition primarily comprises one-dimensional convolution, RepVGG block [37], and dropout [38]. Among these components, the RepVGG block finds extensive usage in image processing. However, due to the distinct patterns of motor parameter data compared to images, adaptive modifications are made to the RepVGG block. The dimension and step size of the convolution kernel within the RepVGG block are adjusted to enhance its suitability for motor parameter data.

### 2.1. Network Model Architecture

The method is to add the RepVGG blocks to the convolutional neural network according to the motor data pattern and make changes to the RepVGG blocks suitable for the parameter data of SRM. The network model is shown in Figure 2.

These data are input into the network, and after being processed and added separately by the two branches of the first layer, the data are transferred to the stacked RepVGG block. The filter size in the RepVGG block is 3 × 1, allowing the filter to capture the data features. The filter stride is set to 2 to reduce redundant feature extraction. Dropout is added before the network’s output layer to select advanced features and mitigate overfitting. All convolutional layers utilize the ReLU (Rectified Linear Unit) activation function and are optimized using the Adam algorithm. In comparison to existing CNN network structures, the network architecture employed in this study demonstrates superior accuracy in predicting motor parameters. This particular network structure is well-suited for extracting deep features related to motor parameters and extracting valuable information from abstract data.

### 2.2. Convolutional Neural Network

Convolutional neural networks (CNNs) were initially proposed for image processing and have achieved impressive results in performing image classification tasks due to their characteristics, such as shared weights and local perception. CNNs have also shown remarkable success in natural language processing, speech recognition, and other fields. The convolutional layer convolves filters with input data to generate features, which are then extracted through pooling operations and learned by stacking filters. While two-dimensional (2D) convolution is commonly used in processing image data, this article uses one-dimensional (1D) convolution to build the model because of the nature of the data. Figure 3 illustrates the process of 1D convolution operation.

The input sequence data entered are $X=[x_{1},x_{2},\cdots ,x_{n}]$, and n represents the length of the X sequence. In convolution operations, the length and number of filters can be defined. For example, the filter length in Figure 3 is 3, the number is 1, and the sequence convolution operation is performed. The calculation formula is as follows:(1)$$y_{1}=w_{1}\cdot x_{1}+w_{2}{\cdot x}_{2}+w_{3}\cdot x_{3}$$

The generated new sequence y extracts feature through convolution operation and downsampling processing, and through the superposition of the convolutional layer and sampling layer, the network continuously extracts and learns important features and finally obtains the model.

In general, the greater the number of filters and layers of convolutional layers and the higher the accuracy of prediction, the greater the computational burden, so it is necessary to weigh it in conjunction with actual research.

### 2.3. RepVGG Network

The RepVGG network has a high inference speed and is suitable for predicting the thrust of SRM. In this study, the lightweight configuration of the RepVGG network was carried out. The attention mechanism was added following the backbone structure of the RepVGG network, further improving the network’s performance.

RepVGG is a network modified from the VGG network [39]. This study mainly consists of 3 × 1 convolution and ReLU (Rectified Linear Unit). A multi-branch model is used during training; a single-way model is used during inference, and the transition is carried out via structural re-parameterization between training and inference. RepVGG adopts the simple main body of the VGG network; at the same time, the residual structure of the ResNet network makes the RepVGG network integrate the advantages of the VGG network and ResNet network.

This model employs a multi-branching structure, and parallel multi-branching generally enhances the model’s characterization ability. The overall model structure is simple, which separates model training from model inference. This separation facilitates accelerated inference and makes the model faster and more accurate. Additionally, the attention mechanism SENet [40] is incorporated into the RepVGG network to capture more detailed information during each round of training, thereby improving the model’s accuracy.

In the fusion process of convolution and BN (Batch Normalization), the formula of the convolutional layer is defined as follows:(2)$$\mathrm{Conv}\left(x\right)=\mathrm{Wx}+b$$
where W is the weight in the filter, and b is the bias, and the bias (b) of training is not considered in this method, so the formula is changed to
(3)$$\mathrm{Conv}\left(x\right)=\mathrm{Wx}$$

The BN layer formula is defined as
(4)$$\mathrm{BN}\left(x\right)=\gamma •\frac{x-\mu }{\sqrt{\sigma^{2}+\varepsilon }}+\beta$$
where μ is the mean; $\sigma^{2}$ is the variance; γ is the contraction coefficient; β is the translation coefficient; ε is a small number to avoid the case of zero denominator. μ and $\sigma^{2}$ are statistically obtained during the training process, and γ and β are obtained by training learning.

Substituting Equation (3) into Equation (4) is obtained.
(5)$$\mathrm{BN}\left(\mathrm{Conv}\left(x\right)\right)=\gamma •\frac{\mathrm{wx}-\mu }{\sqrt{\sigma^{2}+\varepsilon }}+\beta$$

After simplification, the following equation is obtained, which is denoted as
(6)$$\mathrm{BN}\left(\mathrm{Conv}\left(x\right)\right)=\frac{\gamma •w}{\sqrt{\sigma^{2}+\varepsilon }}•x+\beta -\frac{\gamma •\mu }{\sqrt{\sigma^{2}+\varepsilon }}$$
where $\frac{\gamma •w}{\sqrt{\sigma^{2}+\varepsilon }}$ is equivalent to the weight of the new convolution, and $\beta -\frac{\gamma •\mu }{\sqrt{\sigma^{2}+\varepsilon }}$ is equivalent to the bias.

### 2.4. Dropout

Dropout is a method commonly employed in the training of deep learning models, where a portion of the neurons in a neural network are temporarily excluded during the training process. This exclusion is performed randomly, resulting in a subset of neurons being utilized for forward propagation and backpropagation. By selectively deactivating neurons in each training session, dropout effectively reduces the interdependence among them, thereby enhancing the network’s ability to generalize.

The primary goal of dropout is to address the issue of overfitting, which arises when a model becomes too focused on the training data and fails to perform well on unseen test data. By preventing neurons from relying too heavily on specific features or patterns, dropout encourages the network to extract more diverse and robust representations from the available data.

In our specific study, we encountered a limitation in terms of the dataset we had at hand. We collected data from ground trial tests of solid motors, but due to the high costs and time associated with conducting such tests, our dataset is relatively small. Consequently, there is a heightened risk of overfitting when working with this limited amount of data.

To address this challenge and improve the resilience of our model, we decided to incorporate the dropout technique. By randomly deactivating a fraction of neurons during training, we aim to reduce the risk of overfitting and improve the generalization capability of our model. This approach is particularly valuable when dealing with small datasets, as it helps prevent the model from relying too heavily on specific patterns or features extracted from a limited sample.

### 2.5. Cross-Modality Data Prediction Workflow

The network model prediction process is divided into three stages: data preparation; model training; and model testing. The flow chart is shown in Figure 4. In the data preparation stage, the data of the pressure-time and thrust-time sensors during the SRM ground test are preprocessed into one-dimensional data and then divided into training sets and test sets; each sample has the same length. In the model training phase, a network model is built, and the network configuration is determined, including parameters such as the size and number of filters and the number of network layers. The network is trained using the training samples, and during the forward pass, the loss function is computed. The weight parameters are optimized and updated through backpropagation; the Adam optimization algorithm is used in the batch process; the learning rate is changed for training, and the training is completed after all epochs are completed. In the model testing stage, the test data are used for testing; the thrust data are output by inputting the pressure data, and the output thrust is compared with the real thrust for accuracy evaluation.

## 3. Simulation Research

### 3.1. Data Preparation

The data used in this study were obtained from ground test experiments of SRM, where professional sensors were used to sample the combustion chamber pressure and motor thrust, and the collected data were calibrated. However, when collecting data from different models of solid motors with varying operating conditions, the data were collected for different lengths of time, with varying frequencies and numbers of data points. These differences were not suitable for deep learning model training. To facilitate subsequent model learning, the time lengths corresponding to the thrust sequence data and pressure sequence data of the solid motor in the effective operating section were normalized so that the time range was between 0 and 1.

During solid motor testing, the thrust magnitude rises rapidly from zero during the ignition process [41]. The thrust of the entire process can be divided into primary thrust and secondary thrust [42,43]. Secondary thrust is primarily used for aircraft cruising, and the thrust magnitude is generally stable. To facilitate the training and testing of subsequent deep learning models, the thrust data and effective working section of the pressure data are intercepted to prepare samples. Given the high accuracy of pressure data in current measurements, this study utilizes combustion chamber pressure data for end-to-end prediction of thrust data. Based on data analysis and motor operating time, the first 200 points of the entire process are used to prepare samples. Data cropping is shown in Figure 5.

To test the predictive effect of this model, the mean squared error (MSE) and mean absolute error (MAE) were used as the evaluation functions of this model.
(7)MSE=1n∑i=1nyi−yi^2
(8)MAE=1n∑i=1nyi−yi^

### 3.2. Simulation Prediction Results

We present 14 sets of the real data used for training and the corresponding predicted data by downscaling them into a single point in the graph, where each point represents a data line. The distance between data points reflects their relative distance in two dimensions, and two data lines that are very close together in two dimensions will also be close together in the graph. Conversely, if the two data lines are far apart in two dimension spaces, they will also be farther apart in the graph.

As shown in Figure 6, the overall predicted data on the training set are closer to the real data, indicating that the model can effectively learn unknown features and predict desired results during training. The model is well-trained, with no signs of underfitting during training.

The predicted result of the lightweight RepVGG method is shown in Figure 7. By observing Figure 7, we can see that the prediction results of this method in the smoothing section of the data (about 75–200) are very close to the real data. Where the data fluctuate, the prediction results are somewhat different from the real data, but the overall effect is ideal.

### 3.3. Simulation Comparison and Analysis

In this section, the predictive performance of the lightweight RepVGG method is demonstrated by experimental methods, and the superiority of the lightweight RepVGG method is verified by comparing it with other popular neural network structures. The accuracy of this method was demonstrated by comparing the predicted results with the SRM ground trial test data. The experimental method shows the influence of different training sample numbers, filter sizes, block numbers, and window sizes on the prediction performance of the lightweight RepVGG method.

#### 3.3.1. Comparing Network Methods

This study proposes deep learning architecture as an accurate and efficient prediction method. To show the superiority of this approach, this paper compares this network architecture with other well-known network architectures. These include the deep neural network (DNN) [44], long short-term memory recurrent neural network (LSTM) [45], Visual Geometry Group (VGG), residual neural network (ResNet) [46], and other different prediction methods for the evaluation of the prediction results.

The contrasting deep neural network contains three hidden layers, and the number of neurons in the hidden layer is 50, 50, and 100, respectively. The data in this study were measured by SRM ground tests. Due to limited data, the profound network architecture is difficult to train effectively and easily leads to overfitting, so the number of hidden layers and neurons in the model cannot be large. To reduce overfitting, dropout is applied to the last hidden layer.

LSTM is a better network than RNN. RNN can handle specific short-term dependencies, but when the sequence is long, the gradient at the back of the sequence is difficult to backpropagate, and there is a problem of gradient disappearance and explosion, and LSTM can solve this problem. This paper will use a more powerful LSTM instead of traditional RNN as a comparison for this experiment. After testing, two LSTM layers and one fully connected layer were finally used for this experiment, and dropout was also applied to the network to improve the overfitting phenomenon.

ResNet is an excellent network for processing images. In this study, the network configuration of ResNet34 was tried as a comparison model, but it was found that the effect could have been better due to the profound number of network layers. The network can be improved by trying to find how the 12-layer ResNet network works best; dropout is also added to the network.

One of the reasons why VGG network architecture was chosen as an experimental comparison in this study is because RepVGG is an improvement based on VGG, and comparing the two shows the superiority of RepVGG to a certain extent. Of course, for fairness, VGG and RepVGG are identical in parameter design and have the same number of network layers.

In all the comparison methods in this study, the model’s input layer and output layer are the same as the lightweight RepVGG method, and the model parameter update uses backpropagation using the Adam optimization algorithm. Epochs are the same and use MSE as a loss function. All networks have also added dropout to improve network overfitting.

The batch size explicitly influences the network’s performance, and the batch size of 14 samples has the best effect. In the method of variable learning rate employed, the first 500 epochs of the training process are set to 0.001 to optimize the network quickly and then use the learning rate of 0.00005 for accurate convergence. Finally, the test data are input into the trained network for evaluation, and the accuracy is evaluated against the real data.

#### 3.3.2. Analysis of Comparative Results

By integrating the results predicted by different methods, the results are shown in Figure 8. The red line in the left Figure in Figure 8 is the prediction result of the method used in this study; the black line is the real experiment data, and the remaining color line is the prediction result of other methods. Through comparison, we find that the prediction results of all methods are very close to the accurate data in the rising and smooth segments of the data, and all methods have perfect prediction results in these two parts. The main reason is that the data features in the rising and smooth segments are simple, and the simple features can be quickly learned by the deep learning model, so all the methods in these two parts have good prediction results. However, different methods show other differences where the data have an apparent fluctuation range (20–50). Zooming in on the fluctuation section shows that the red line is the closest to the accurate data. Although there is some deviation in the first half of the section, the latter is closer to the accurate data. However, the prediction results of other methods in the fluctuation period are significantly different from the accurate data, and the biggest difference is the green line. In order to avoid the chance of the experiment, we replaced the experiment data and verified them again. The results are shown in the right picture in Figure 8. Only the test data are replaced when all configurations are identical. Through observation, we can also see that the red line is closest to the black line compared with the lines of other colors. Although the lines of other colors are close to the actual data trend in the overall trend, the red line is closer to the accurate data, which means that the prediction is more accurate. MSE evaluates the prediction results of different methods shown in Figure 9. It is easy to see that several of these methods perform well in the test set; the lightweight RepVGG method performs the best; LSTM, ResNet, and VGG perform equally, and DNN performs the worst.

#### 3.3.3. Effects of the Number of Training Samples and Window Size

In this part, we study the effect of experimental parameters on the model test results. Figure 10 shows the effect of the number of training samples on the model test effect. The data used in this study come from the ground experiment of the SRM. Due to the high cost and long experiment period of the ground experiment of the rocket motor, the data are limited. We divide the data into training sets and test sets under limited data. We can observe that the more there are training samples, the smaller the corresponding MSE and MAE, which means that if there are more training samples, the more accurate is the model prediction. Using more training samples will make the model fully trained to achieve better results. Another factor affecting the model’s prediction effect is the window size. The window size’s impact on the model’s prediction effect is shown in Figure 11. In the data processing part, it is proposed to intercept the first 200 data points of the whole data segment to prepare samples, so the maximum window is 200 when setting the window size. Through observation, we find that the larger the window size is set, the smaller the corresponding MSE and MAE are, and the closer the prediction result of the model is to the actual data value. More oversized windows can cover more data information and contain more data features.

#### 3.3.4. Effects of Number of Blocks and Filter Sizes

The size of the filter and the number of blocks also have an effect on the prediction performance of the model, and we experimentally documented model errors when we tested them with different filter sizes and different numbers of blocks. Figure 12 shows that the training time increases linearly with the number of blocks, which increases the computational pressure. The model’s performance with more blocks on the test set differs. The reason may be that with the increase in the number of blocks, the training parameters of the model also increase, and the training set is a small data set, resulting in an easy overfitting [47]. As a result, models with more blocks on the test set perform poorly. In a comprehensive comparison, it is found that the model with five blocks has good performance, which is used as the predictive model of the lightweight RepVGG method.

The lightweight RepVGG method investigates the impact of the convolutional filter size on network performance. An experimental method is employed to compare the effect of different filter sizes on test accuracy, and the model training time is also measured. Figure 13 illustrates that using a filter size of 3 results in the smallest MSE and MAE on the test, indicating that the model performs best with a filter size of 3. Additionally, the training time increases with larger filter sizes while prediction accuracy decreases. Smaller convolutional kernels can extract fine features, enabling the model to learn abstract features. Moreover, smaller kernels require fewer training parameters, reducing model complexity and accelerating training.

## 4. Conclusions

In response to the problem of scarcity of thrust data and large measurement errors of solid rocket motors, this paper proposes a lightweight RepVGG-based method for cross-modality data prediction of SRM. By processing the raw data, a model suitable for solid rocket motor data is built with the aim of predicting thrust data by inputting the pressure data of solid rocket motors. By training the model sufficiently, this method achieves good results on the test set. The predicted results show small errors and are close to the actual data, particularly in the rising and smoothing sections.

Compared with other popular methods, this method shows the accuracy and effectiveness of prediction. The influence of the number of training samples on the model test results is studied. It is found that the prediction effect is best when the number of training samples is 14. At the same time, the influence of the window size on the model test effect is also analyzed. Through experiments, it is found that when the window is set to 200, the prediction effect of the model is the best. Through experiments, it is found that the number of model blocks has an important impact on the test results. Correct selection of the number of blocks has an important impact on model testing. During data preprocessing, the original data are large, and the sampling frequency of the data is different. The first 200 data points are directly intercepted during processing to prepare samples.

Although the method in this paper has achieved good prediction results, the predicted data are also close to the real rocket motor experiment data. However, the training data set is smaller due to the scarcity of rocket motor data. The model trained on small data sets is prone to overfitting. Although the overfitting can be alleviated by adding dropout, it will still affect the test of the model. In addition, the sample is prepared by directly intercepting the first 200 data points, which also has certain limitations. Therefore, future work can employ data interpolation methods to insert new data points between existing ones, thereby increasing the density and capturing more details in the data.

## Figures and Tables

**Figure 1 sensors-23-09165-f001:**
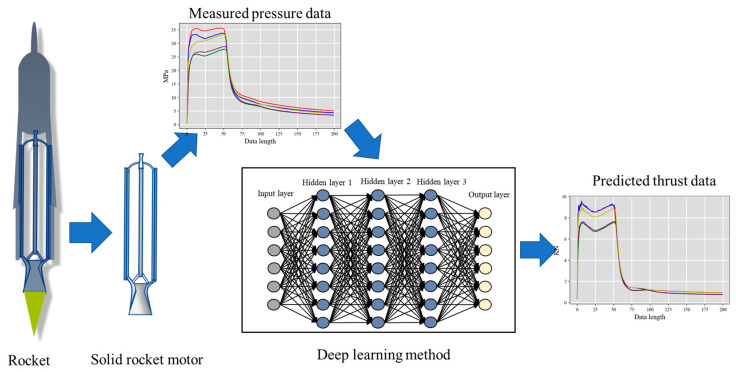
The workflow illustration.

**Figure 2 sensors-23-09165-f002:**
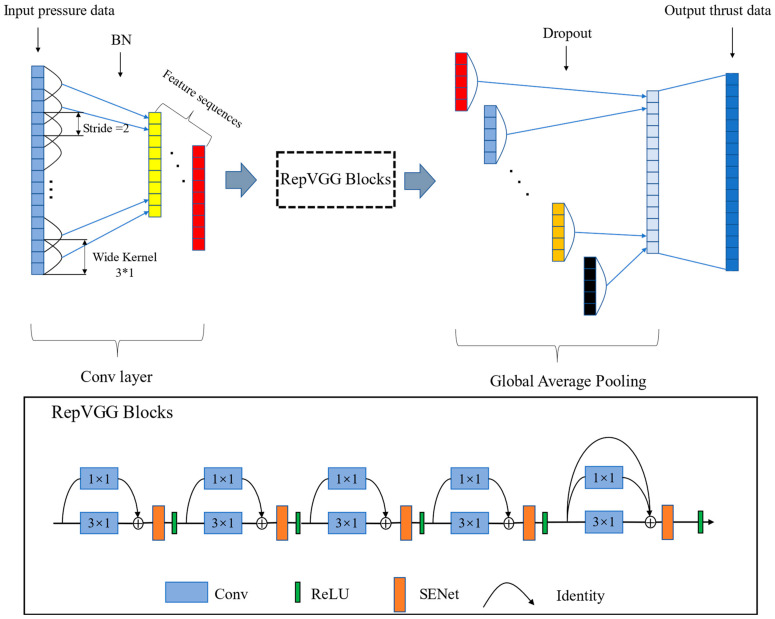
Illustration for the network model of the lightweight RepVGG method.

**Figure 3 sensors-23-09165-f003:**
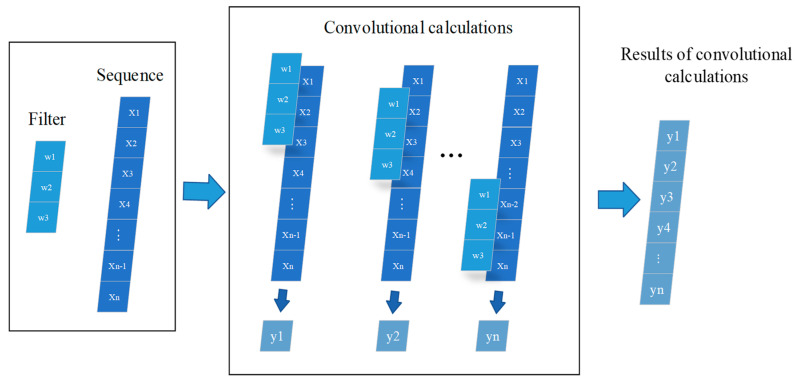
Figure for 1D CNN operation.

**Figure 4 sensors-23-09165-f004:**
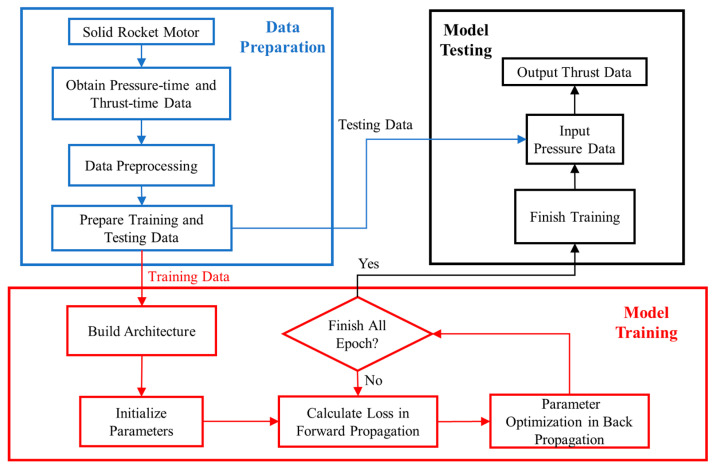
Flow chart of the lightweight RepVGG method for prognostics.

**Figure 5 sensors-23-09165-f005:**
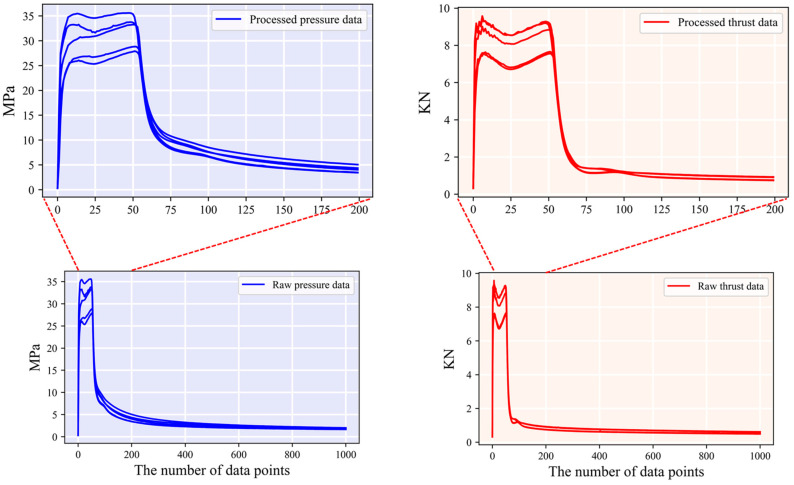
Crop raw pressure and thrust data.

**Figure 6 sensors-23-09165-f006:**
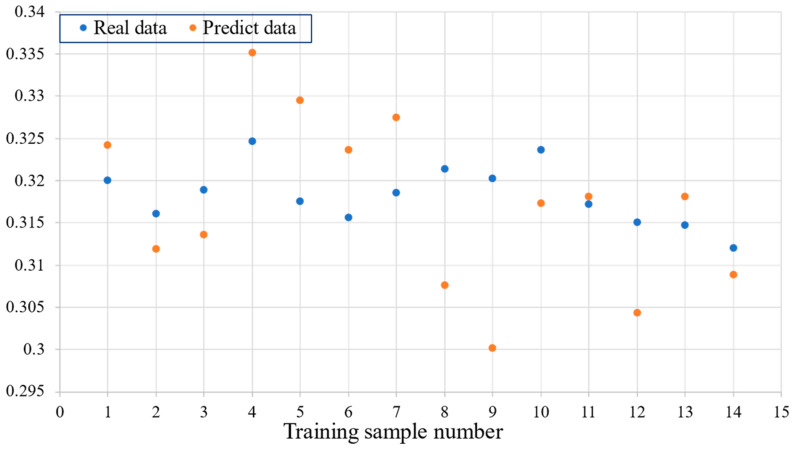
Performance of the lightweight RepVGG method on the training set.

**Figure 7 sensors-23-09165-f007:**
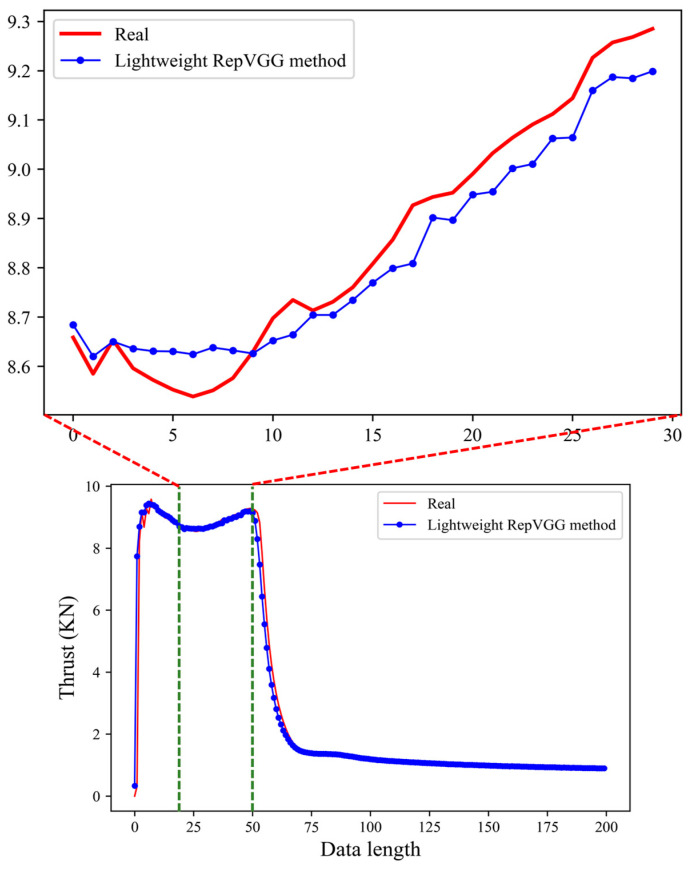
Thrust prediction results of the lightweight RepVGG method.

**Figure 8 sensors-23-09165-f008:**
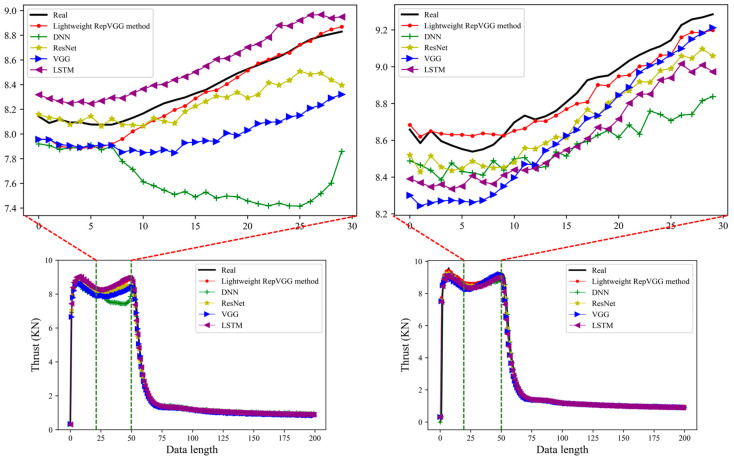
The test results of the lightweight RepVGG method are compared with the other four methods.

**Figure 9 sensors-23-09165-f009:**
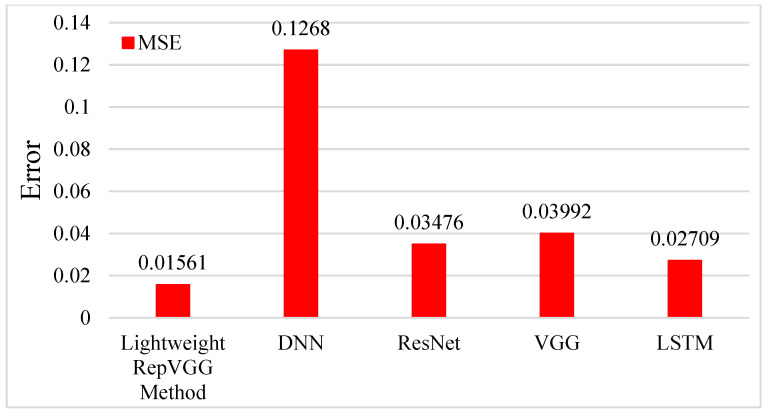
The performance of different methods on the test set.

**Figure 10 sensors-23-09165-f010:**
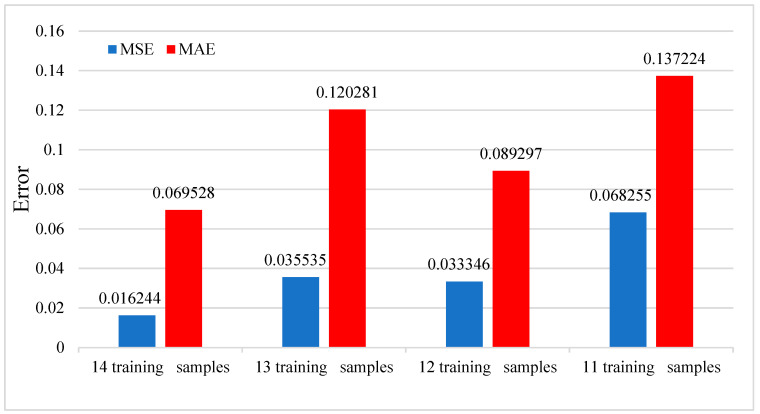
The effect of the number of training samples on the predictive performance of the model on test set.

**Figure 11 sensors-23-09165-f011:**
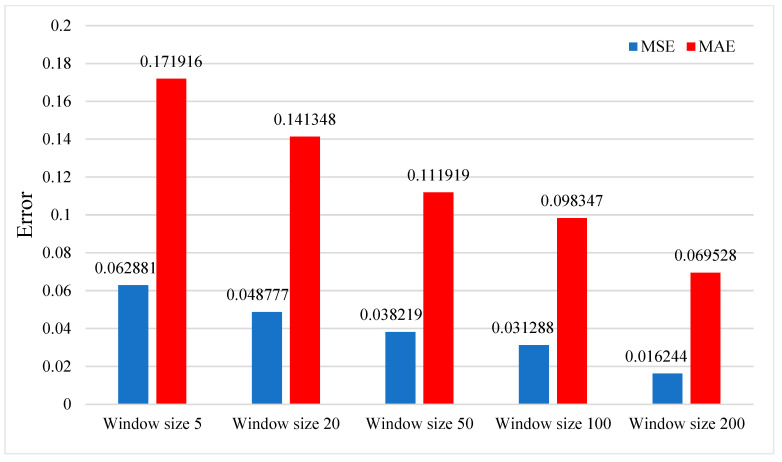
The effect of window size on the predictive performance of the model on the test set.

**Figure 12 sensors-23-09165-f012:**
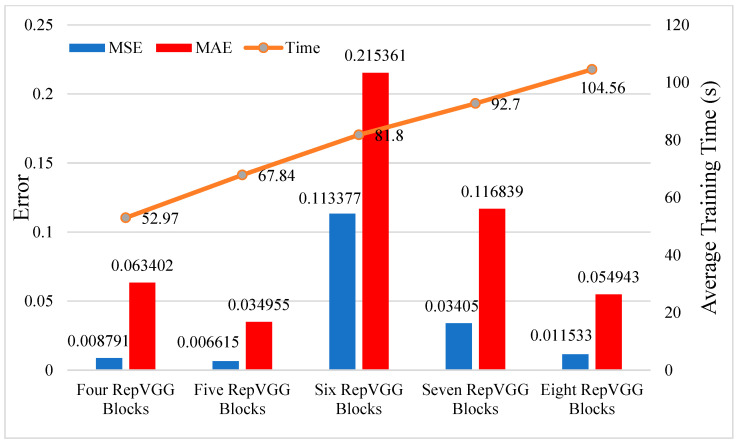
The effect of the number of blocks on prediction performance and calculation time in the lightweight RepVGG method.

**Figure 13 sensors-23-09165-f013:**
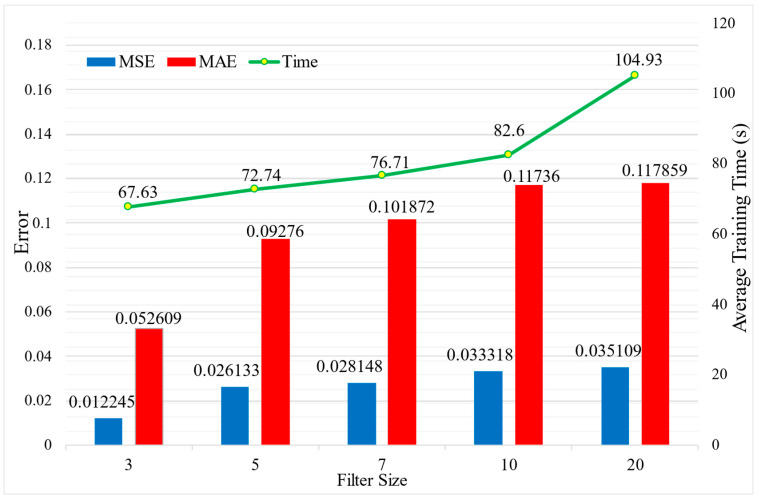
The effect of filter sizes on prediction performance and computation time in the lightweight RepVGG method.

## Data Availability

No new data were created. The data used are confidential.

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
