# Peer review of "Lightweight RepVGG-Based Cross-Modality Data Prediction Method for Solid Rocket Motors"

_sensors, 2023, doi:10.3390/s23229165_

Round 1

Reviewer 1 Report

A Lightweight RepVGG Based cross-modality data prediction method is proposed for SRM. An end-to-end data prediction framework is established through transforming data across different modalities. A novel RepVGG deep neural network architecture is built, which is able to automatically learn features from raw data and predict new time-series data of different modality. The research work reported is interesting in the community. Some suggestions are listed below to improve the manuscript's quality (major revision):

1. The manuscript's motivations should be further highlighted in the manuscript, e.g., what problems did the previous works exist? How to solve these problems? The authors may consider analyzing the problems of the previous works and how to address these problems with the proposed method. Please explain that.

2. The research gaps in the abstract and introduction should be clearly expressed. Please rewrite this part.

3. The authors must clearly explain the difference(s) between the proposed method and similar works in the introduction. The authors should further highlight the manuscript's innovations and contributions.

4. In Section 2.3 of RepVGG network, how to determine the structure? Please provide.

5. One key background of this manuscript is the advanced techniques. Thus, the Introduction and/or related work section could be extended and incorporates additional discussions on the topics of advanced techniques, e.g.,  https://doi.org/10.1109/TR.2022.3180273; http://dx.doi.org/10.1145/3513263; http://dx.doi.org/10.1016/j.marstruc.2022.103338 and so on.

Moderate editing of English language required

Author Response

Thank you so much for your positive comments on this paper. We are greatly appreciated. We have carefully and positively considered all the comments. Extensive revisions have been made in the revised manuscript. The changes are generally presented in this Response Letter, and also highlighted as the red words in the texts. 

Reviewer 2 Report

This paper proposes a Lightweight RepVGG Based cross-modality data prediction method, which is able to automatically learn features from raw data and predict new time-series data of different modality. Experimental results prove the effectiveness and superiority of this method. However, there still exist some problems need to be revised, which are presented as follows:

1. The contribution and innovation of the manuscript should be clarified clearly in abstract and introduction.

2. The literature review of this paper should be enriched to enhance the introduction part of the manuscript, such as:

10.1088/1361-6501/ac9543

3. The quality of the figures should be higher, such as Figure 1, Figure 3, etc.

4. Spelling and grammatical errors should be checked carefully so that the language of the paper can be more standardized and readers can have a good reading experience.

5. How is the data collected? How is the data preprocessed? The author can enrich some relevant content to make it easier for readers to understand.

6. The conclusion section mentions the limitations brought by the data. In response to this issue, the author can briefly introduce future work and research directions.

Spelling and grammatical errors should be checked carefully so that the language of the paper can be more standardized and readers can have a good reading experience.

Author Response

(The authors gave the same response as above.)

Reviewer 3 Report

This paper proposes Lightweight RepVGG Based Cross-Modality Data Prediction Method for Solid Rocket Motor. In general, this paper is well presented. The idea is interesting, and the method is novel. The application on the solid rocket motor shows its effectiveness. This paper can be published. However, the following issues can be further considered.

1. More background and motivation of this study can be added, in case the readers are not very familiar with the topic.

2. The descriptions of the well known knowledge can be properly reduced.

3. Why introducing the cross modality data prediction method? What is the major benefits compared with traditional methods?

4. Some more recent related works on this topic should be reviewed, to highlight the new contributions of this study.

5. The quality of the figures should be improved.

Minor editing of English language required

Author Response

(The authors gave the same response as above.)

Round 2

Reviewer 1 Report

ok